# Increasing Al-Pair Abundance in SSZ-13 Zeolite via Zeolite Synthesis in the Presence of Alkaline Earth Metal Hydroxide Produces Hydrothermally Stable Co-, Cu- and Pd-SSZ-13 Materials

Konstantin Khivantsev [1,*,†], Miroslaw A. Derewinski [1,2,*,†], Libor Kovarik [1], Mark Bowden [1], Xiaohong Shari Li [1], Nicholas R. Jaegers [1], Daria Boglaienko [1], Xavier I. Pereira-Hernandez [1], Carolyn Pearce [1], Yong Wang [1,3] and Janos Szanyi [1,*]

[1] Pacific Northwest National Laboratory, Richland, WA 99352, USA; libor.kovarik@pnnl.gov (L.K.); mark.bowden@pnnl.gov (M.B.); xiaohong.li@pnnl.gov (X.S.L.); njaegers@berkeley.edu (N.R.J.); daria.boglaienko@pnnl.gov (D.B.); vruatsa@gmail.com (X.I.P.-H.); carolyn.pearce@pnnl.gov (C.P.); wang42@wsu.edu (Y.W.)

[2] J. Haber Institute of Catalysis and Surface Chemistry, Polish Academy of Sciences, 30-239 Krakow, Poland

[3] The Gene and Linda Voiland School of Chemical Engineering and Bioengineering, Washington State University, Pullman, WA 99164, USA

[*] Correspondence: konstantin.khivantsev@pnnl.gov (K.K.); ncderewi@cyf-kr.edu.pl (M.A.D.); janos.szanyi@pnnl.gov (J.S.)

[†] These authors contributed equally to this work.

**Abstract:** Replacing alkaline for alkaline-earth metal hydroxide in the synthesis gel during the synthesis of siliceous SSZ-13 zeolite (Si/Al~10) yields SSZ-13 with novel, advantageous properties. Its $NH_4$-form ion-exchanges higher amount of isolated divalent M(II) ions than the conventional one: this is the consequence of an increased number of Al pairs in the structure induced by the +2 charge of Sr(II) cations in the synthesis gel that force two charge-compensating $AlO_4^-$ motives to reside closer together. We characterize the +2 state of Co(II) ions in these materials with infra-red spectroscopy and X-ray absorption spectroscopy measurements and show their utility for NOx pollutant adsorption from ambient air: the ones derived from SSZ-13 with higher Al pair content contain more isolated cobalt(II) and, thus, perform better as ambient-air NOx adsorbers. Notably, Co(II)/SSZ-13 with an increased number of Al pairs is significantly more hydrothermally stable than its NaOH-derived analogue. Loading Pd(II) into Co-SSZ-13(Sr) produces an active NOx adsorber (PNA) material that can be used for NOx adsorption from simulated diesel engine exhaust. The critical issue for these applications is hydrothermal stability of Pd-zeolites. Pd/SSZ-13 synthesized in the presence of $Sr(OH)_2$ does not lose its PNA capacity after extremely harsh aging at 850 and 900 °C (10 h in 10% $H_2O$/air flow) and loses only ~55% capacity after hydrothermal aging at 930 °C. This can be extended to other divalent metals for catalytic applications, such as copper: we show that Cu/SSZ-13 catalyst can survive hydrothermal aging at 920 °C without losing its catalytic properties, metal dispersion and crystalline structure. Thus, we provide a new, simple, and scalable strategy for making remarkably (hydro)thermally stable metal-zeolite materials/catalysts with a number of useful applications.

**Keywords:** divalent metals in zeolites; copper palladium cobalt ions in zeolite SSZ-13; catalysts and adsorbers for nitric oxide NOx emissions control; elective catalytic reduction (SCR) and NOx adsorbers

## 1. Introduction

Within the last decade, metal/zeolites system have become the front-liners of environmental catalysis [1,2]. These materials/catalysts have allowed us to tackle the most challenging environmental problems caused by pollutants emitted by vehicles [1–8]. More

specifically, nitric oxide is a free-radical harmful pollutant, even more dangerous than carbon monoxide (CO). Nearly 55% of the global NOx emissions come from vehicles [1,9–11]. Cu/zeolite materials have been successfully commercialized to scrub NOx and turn it into environmentally clean dinitrogen in diesel vehicles [1–7]. Sacrificial ammonia reductant is used, and this catalytic reaction is effective at temperatures starting >170 °C with state-of-the-art catalyst formulations. Significant advances have been achieved in understanding the fundamental nature of catalytic SCR reaction on Cu/SSZ-13 and Cu/BEA zeolitic systems [1–7,12–22]. Although Cu/SSZ-13 catalysts show excellent activity above 200 °C, no catalysts were shown to be active at lower temperatures. Only recently we have discovered that $NO^+$ ions in zeolites (formed via $NO + O_2$ reaction) catalyze direct reduction of $NO^+$ with $NH_3$ to dinitrogen at room temperature and below [19]. The process can be run catalytically when some $NH_3$ can desorb from zeolite [19]. However, higher temperatures require Cu(II) ions presence due to, in part, the ability to form Cu(I)---$NO^+$ site by direct Cu(II) reduction with free-radical NO. Emissions during vehicle cold start, however, remain a challenge. To circumvent this challenge, Pd/zeolite materials (passive NOx adsorbers) have been introduced industrially that adsorb NO strongly as Pd(II)-NO, Pd(I)-NO, Pd(II)(OH)(NO) and Pd(II)(NO)(CO) complexes (depending on Pd speciation in zeolites) at low temperatures (80–130 °C) during cold start and then release them at higher temperatures (>170–180 °C) when downstream SCR catalysts become very active [20,21,23–38]. The main challenge for the application of metal/zeolite materials for industrial vehicle use has always been their hydrothermal stability. They may be exposed in vehicles to harsh steam treatment (during diesel particular filter regeneration, for example) [23–37]. In general, it is desirable to develop materials that can survive hydrothermal aging at temperatures as harsh as 800 °C and higher. For passive NOx adsorbers, the highest known survival temperatures are 750–820 °C [23–37]. For example, we have been able to engineer Pd/BEA crystals that can survive 750 °C hydrothermal treatment without any deterioration [25]. For SSZ-13 materials prepared via the regular method (in the presence of NaOH and N(adamantyl) structure-directing agents), 750 °C hydrothermal aging (HTA) already leads to erosion of some Pd(II) ions and their transformation into nanoparticles on the external zeolite surface, likely through Pd aluminate and $Pd_x(OH)_y$ species formation in the presence of water (with simultaneous framework dealumination) [20,21,29–31]. Therefore, there is an urgent need to develop hydrothermally stable SSZ-13 materials. It is important to note that the Si/Al ratio of zeolite plays an important part in stability: the higher the Si/Al ratio, the more hydrophobic the inner pore of zeolite is (considering no significant number of other defects are present) and the more resistant to steam HTA treatment it is [25,31]. However, as we convincingly showed recently, with an increase of Si/Al ration from 6 to 10–12 to 20–30, hydrophobicity of the pores leads to exclusion of atomically dispersed M(II) species and formation of MOx nanoparticles on the external surface (due to hydrophobicity of the pore and the inability of the aqua hydroxo complexes of metal to get "inside") [24,26,29]. Furthermore, Pd distribution changes with increase of Si/Al ratio: for SSZ-13 with Si/Al ratio 6, Pd was almost completely atomically dispersed and present exclusively as Pd(II) ions held by proximal $AlO_4^-$ pairs [24,26]. For samples with an Si/Al ratio 10–12, around ~80–90% Pd was atomically dispersed and presented as a mixture of Pd(II) and Pd(II)-OH species [24,26]. For samples with higher Si/Al ratios, much less (<30%) Pd was atomically dispersed and presented exclusively as Pd(II)-OH species [24,26]. Pd(II) ions are more active and stable towards PNA. Thus, we chose to focus on the zeolite with Si/Al ~10 (providing higher hydrothermal stability than Si/Al~6) that can still accommodate most of Pd as atomically dispersed species. Furthermore, for the sample with Si/Al ~5–6, most of Al would be present as proximal pairs, most likely, irrespective of synthesis conditions due to a significant abundance of Al. However, for higher Si/Al ratios statistically speaking, fewer Al sites will be proximal and thus it would be harder to create the desired M(II) sites. We hypothesized that we could create more Al paired sites by introducing alkaline earth metals (such as $Sr(OH)_2$ or $Ba(OH)_2$) in the synthesis gel instead of the typically used NaOH. We focused on Sr specifically rather than

Ba, because Ba is extremely toxic (however, we believe barium hydroxide would probably have similar effects as strontium hydroxide). $Sr(OH)_2$ has some issues with solubility at room temperature (it takes some stirring to dissolve it), but at higher temperatures of the typical hydrothermal autoclave synthesis it has excellent solubility. Sr(II) has a charge of +2, and it would require two proximal $AlO_4^-$ units to produce the resulting Sr(II)-form zeolite compared with the 1 $AlO_4^-$ unit required to stabilize Na-form zeolite [3].

## 2. Results and Discussion

### 2.1. Formation of Highly Crystalline SSZ-13 Zeolite during Synthesis in the Presence of Sr(OH)$_2$

The synthesis of the SSZ-13 zeolite in the presence of NaOH is well established. However, replacement of NaOH with $Sr(OH)_2$ has not been known or studied to date. We note that other alkaline-earth metals, such as $Mg(OH)_2$ or $Ca(OH)_2$, may not be suitable for this synthesis as they are poorly soluble both in hot and cold solutions. $Sr(OH)_2$ has good solubility at the temperature of the synthesis and creates the required basic environment during the synthesis: it was, thus, chosen by us. Toxicity issues with using and handling barium hydroxide in our facilities precluded us from studying its use in the synthesis. However, this could be explored in follow-up studies. X-ray diffraction (XRD) patterns of the as-synthesized material show highly crystalline zeolite SSZ-13 and the absence of other phases, which proves that the newly developed synthesis method with NaOH replacements is suitable for producing crystalline SSZ-13 materials. The average crystal size estimated on the basis of XRD for fresh SSZ-13 crystals on the basis of the Scherrer equation ~120 nm (Figure 1).

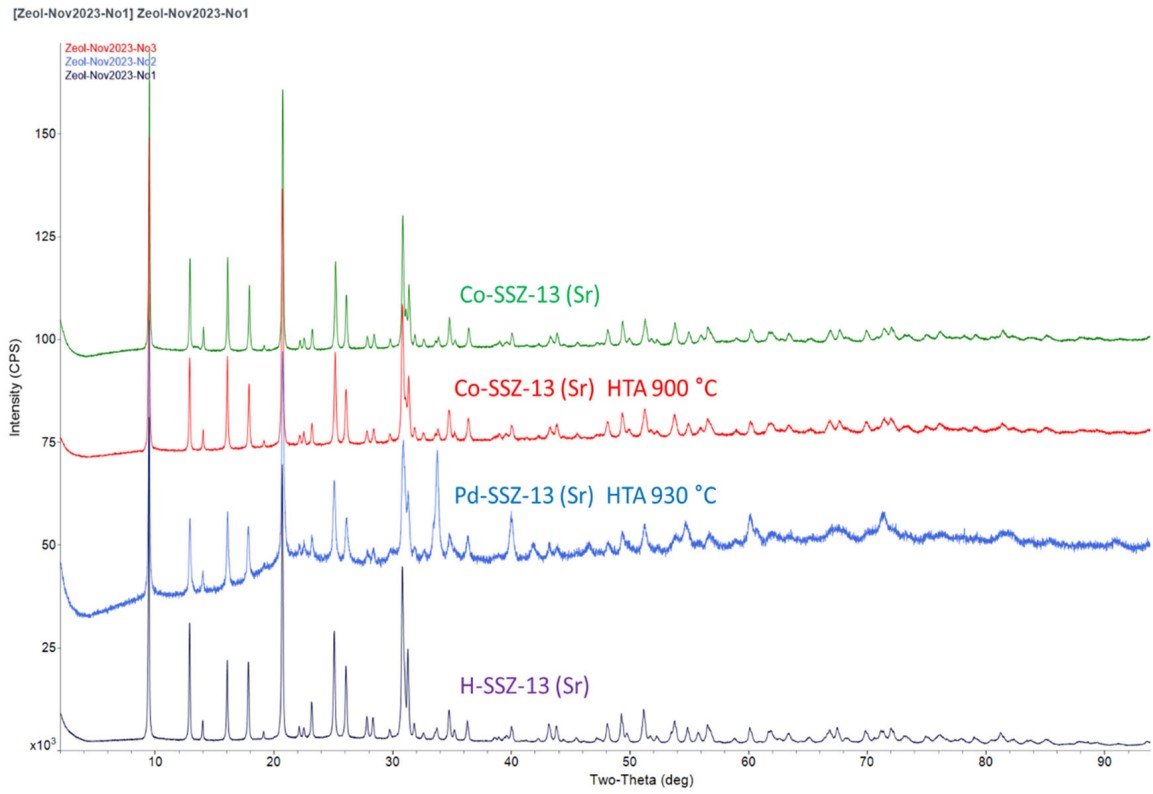

**Figure 1.** XRD patters of various SSZ-13 (Sr) samples: (1) SSZ-13 in the H-form, prepared by thermal decomposition of NH4-SSZ-13 at 550 °C, showing highly crystalline Chabazite material; (2) Pd-containing Co-SSZ-13 (Sr) material (from Figure 2) after extremely harsh hydrothermal aging at 930 °C in 10% air/water steam, showing remarkable preservation of SSZ-13 crystals after such extraordinarily harsh hydrothermal aging; (3) Co-SSZ-13 (Sr) material from Figure 3 after extremely harsh hydrothermal aging at 900 °C: crystallinity is preserved; (4) Co-SSZ-13 (Sr) fresh material.

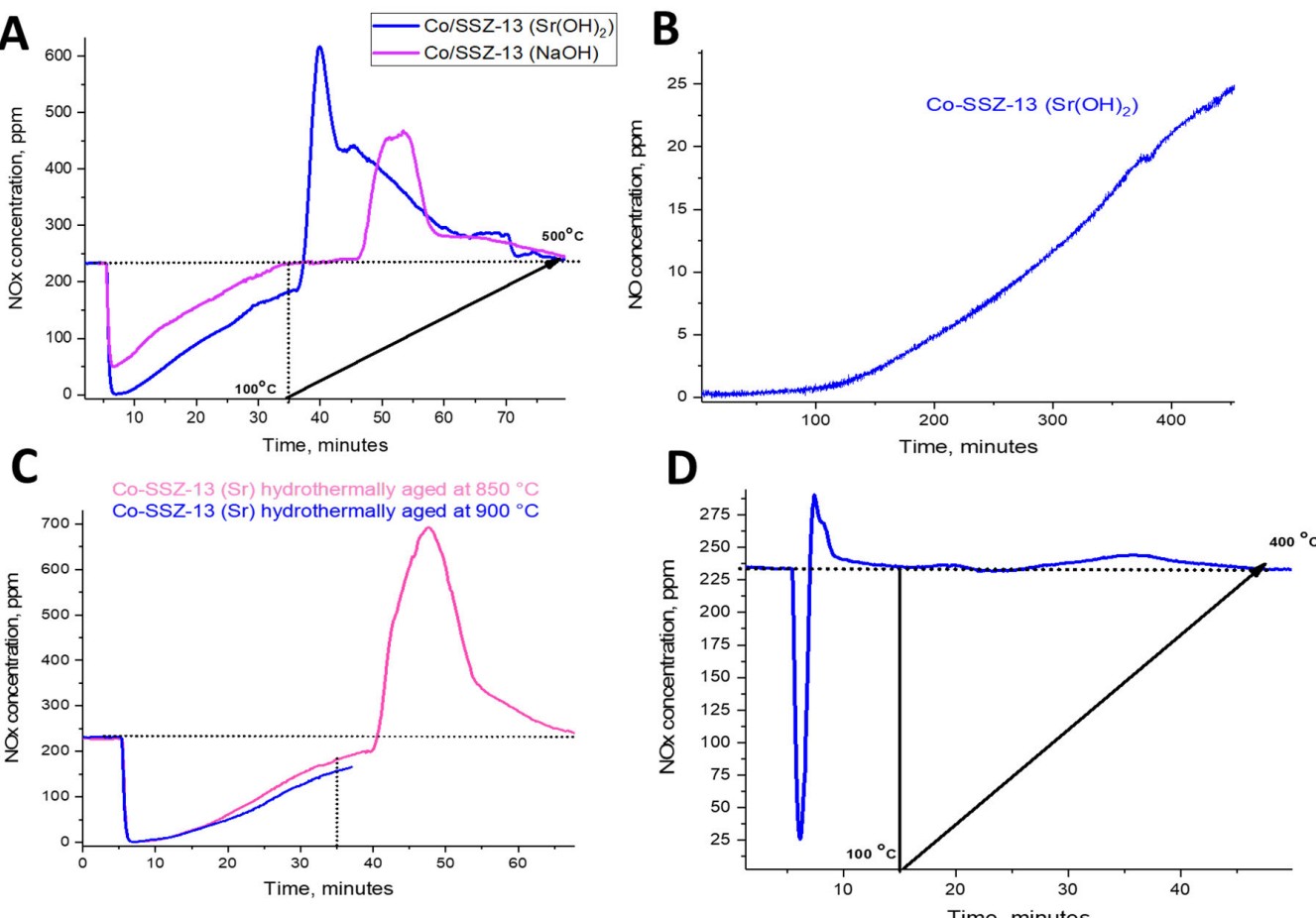

**Figure 2.** (**A**) NOx adsorption on Co-SSZ-13 samples derived from NaOH and Sr(OH)$_2$ synthetic routes at 100 °C for 30 min (after 5 min bypass) followed with TPD (10 °C/min). The feed gas mixture contains approx. 220 ppm of NOx, 14% O$_2$ and 300 ppm H$_2$O. Sample mass 120 mg. Total flow rate 300 sscm/min. (**B**) NOx adsorption on Co-SSZ-13 samples derived from an Sr(OH)$_2$ synthetic route at 23 °C while a flowing gas mixture containing 27 ppm of NO, 20% O$_2$ and 300 ppm H$_2$O (adoption of NO from ambient air keeps the NO level below 2 ppm for 160 min). Sample mass 120 mg. Total flow rate 300 sscm/min. (**C**) NOx adsorption on a Co-SSZ-13 sample derived from an Sr(OH)$_2$ synthetic route at 100 °C for 30 min (after 5 min bypass) followed with TPD (10 °C/min). The feed gas mixture contains 220 ppm of NOx, 14% O$_2$ and 300 ppm H$_2$O. Sample mass 120 mg. Total flow rate 300 sscm/min. The sample was first hydrothermally aged at 850 °C and tested for PNA. Then it was further hydrothermally aged at 900 °C and PNA adsorption capacity was recorded at 100 °C for 30 min (after 5 min bypass). (**D**) NOx adsorption on Co-SSZ-13 samples derived from an Sr(OH)$_2$ synthetic route at 100 °C for 10 min (after 5 min bypass) followed with TPD (10 °C/min). The feed gas mixture contains 220 ppm of NOx, 200 ppm CO, 14% O$_2$ and 3% H$_2$O. Sample mass 120 mg. Total flow rate 300 sscm/min. Note that for vehicle-related applications, adsorption is conducted in the presence of higher amounts of moisture and typical exhaust gases at 100 °C. For adsorption of NOx at ambient pressure and temperature (NO as an air pollutant), typical conditions do not include high water levels.

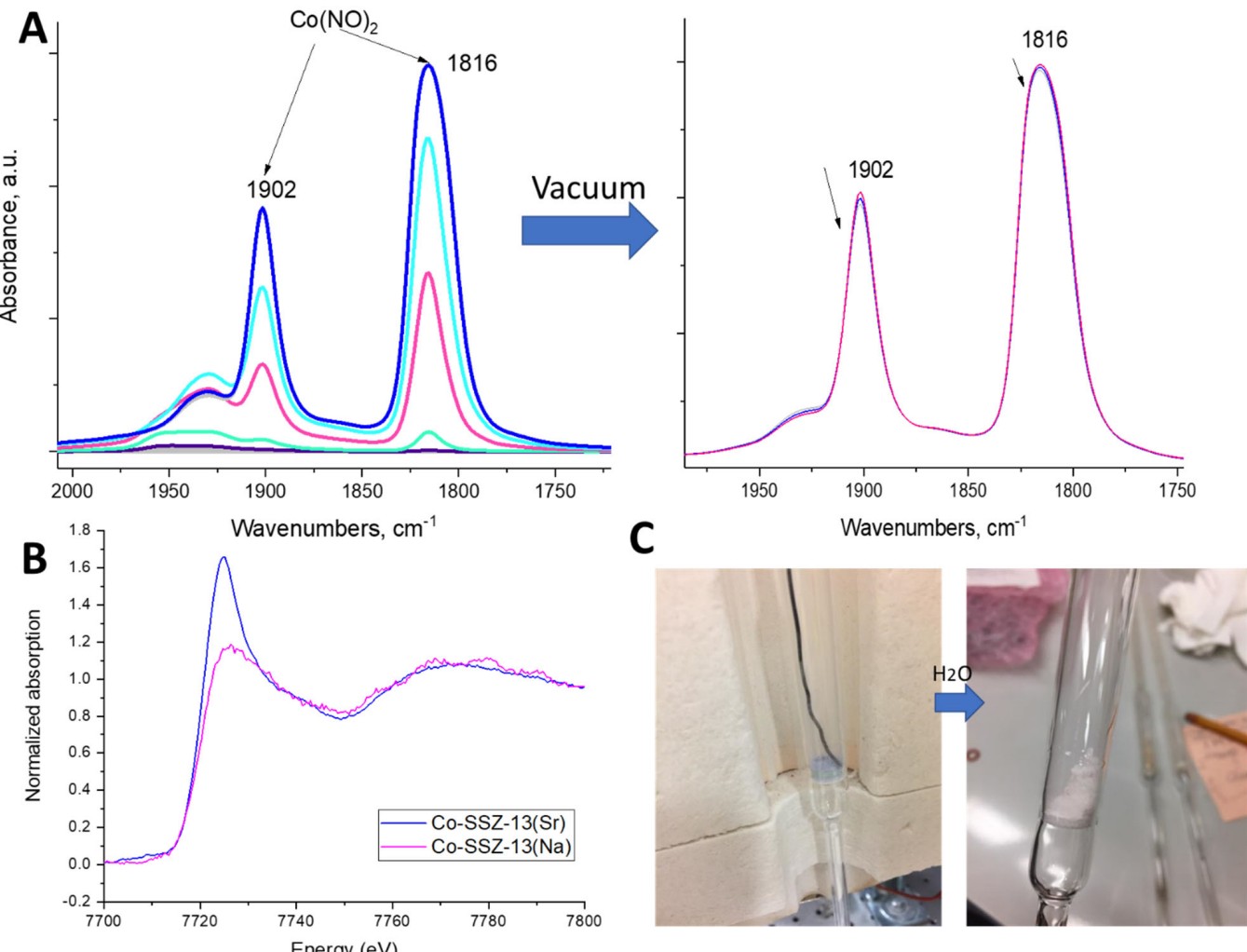

**Figure 3.** (**A**) In-situ FTIR during sequential NO adsorption (2 Torr) at room temperature on activated Co-SSZ-13 (Sr). NO bands at 1902 and 1816 cm$^{-1}$ belong to the Co(II)(NO)$_2$ complex. Evolution of spectra upon NO adsorption is shown in different colors. Minor band at 1930 cm$^{-1}$ belongs to NO adsorbed on Co(III) ions. NO desorption from Co(II)(NO)$_2$ under vacuum (10$^{-6}$ Torr) at 80 °C. Co(II)(NO)$_2$ complex shows resistance to decomposition. (**B**) Co K-edge XANES spectrum for Co-SSZ-13(Na) and Co-SSZ-13(Sr) samples. (**C**) Dehydrated Co-SSZ-13(Na) sample. Blue color is due to the presence of Co(II) cations in a distorted tetrahedral ligand environment. The same sample was exposed to ambient air for 7 days. This sample slowly adsorbs moisture from ambient air forming pink-ish Co(II)/SSZ-13 with hydrated octahedral (or square-pyramidal) Co(II) cations.

We looked at prepared H-SSZ-13 material using HAADF-STEM imaging and EDS mapping (Figure 4). These revealed nicely shaped cubic crystals of zeolite of high purity. EDS shows the presence of Si, Al, O as expected for this zeolite. The Si/Al ratio on the basis of EDS mapping was ~10.

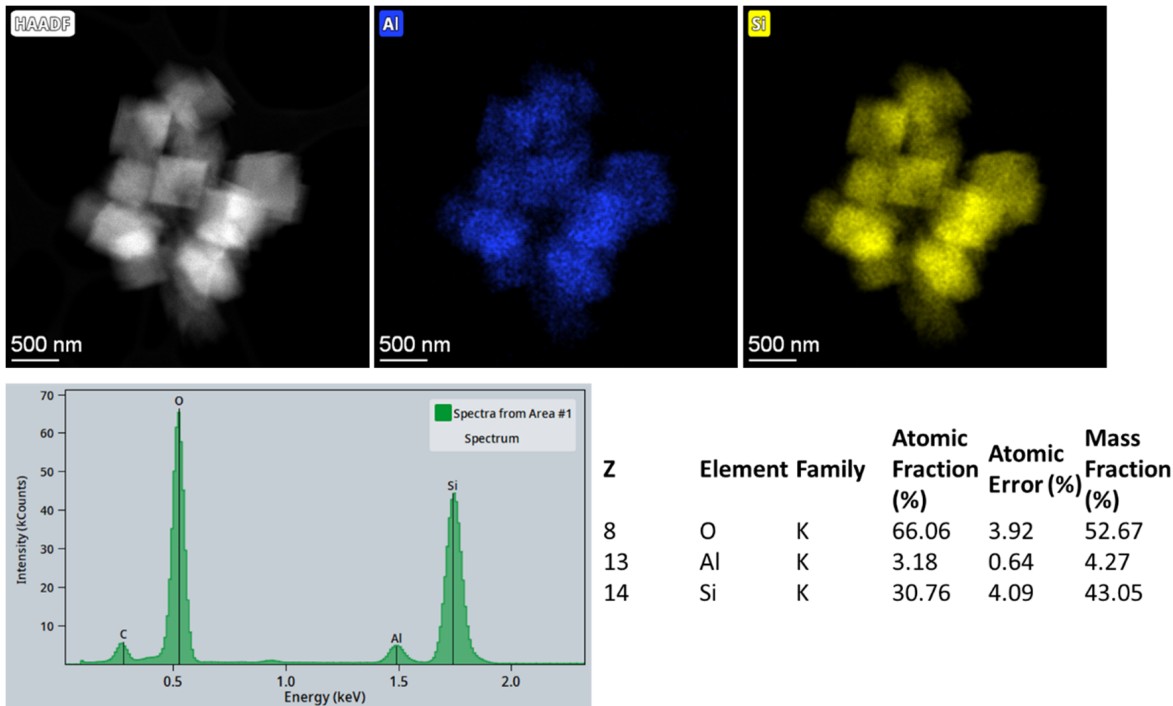

**Figure 4.** Representative HAADF-STEM image of H-SSZ-13 (Sr) crystals and the corresponding EDS mapping of these crystals, showing the presence of Si, Al, O. The corresponding mass and atomic concentrations of Si, Al, O are summarized in the table.

Utilization of EDS also allowed us to look at the variation of Al density along the edge of the crystals vs. the bulk of the crystals (Figure S1). Choosing areas that encompass either the whole crystal (average Si/Al ratio), the crystal periphery/edge (edge Si/Al ratio) or the crystal core (core Si/Al) ratio reveal that the ratios are quite similar throughout the crystal and Al is well distributed and no dramatic gradient can be found (Figure S1 and Table S1). However, we must note that there is a statistically significant presence of slightly lower amounts of Al on the edges, with EDS-determined ratios Si/Al~11 at the edge vs. Si/Al~9.5 at the core.

We also performed BET and pore size measurements for H-SSZ-13 (Sr) and H-SSZ-13 (Na) samples (Figure 5 and Table 1). Both samples show nitrogen adsorption isotherms consistent with a well-defined microporous material with a steep nitrogen uptake at low $P/P_0$ ratios (Figure 6 shows BJH pore distribution during adsorption).

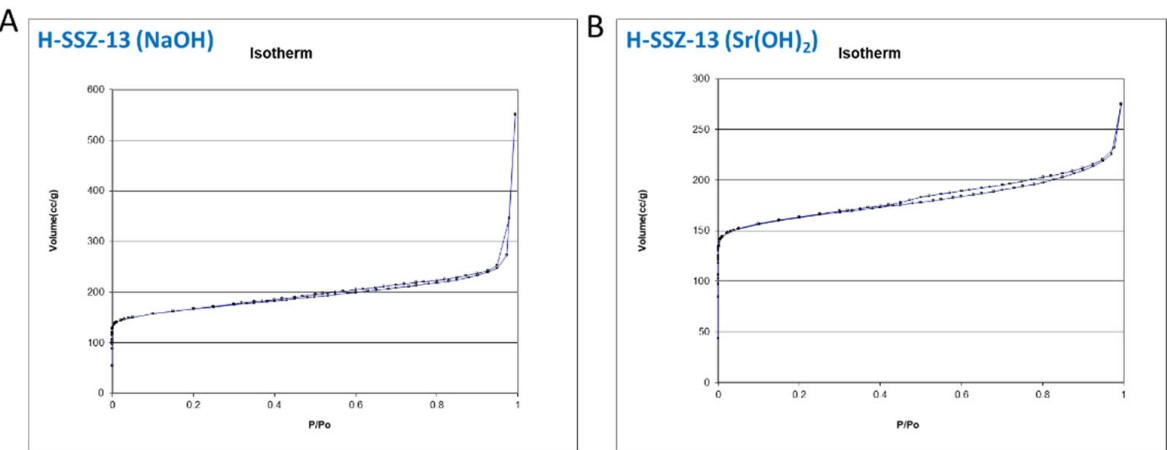

**Figure 5.** Nitrogen BET adsorption isotherms for H-SSZ-13 (Na) and H-SSZ-13 (Sr).

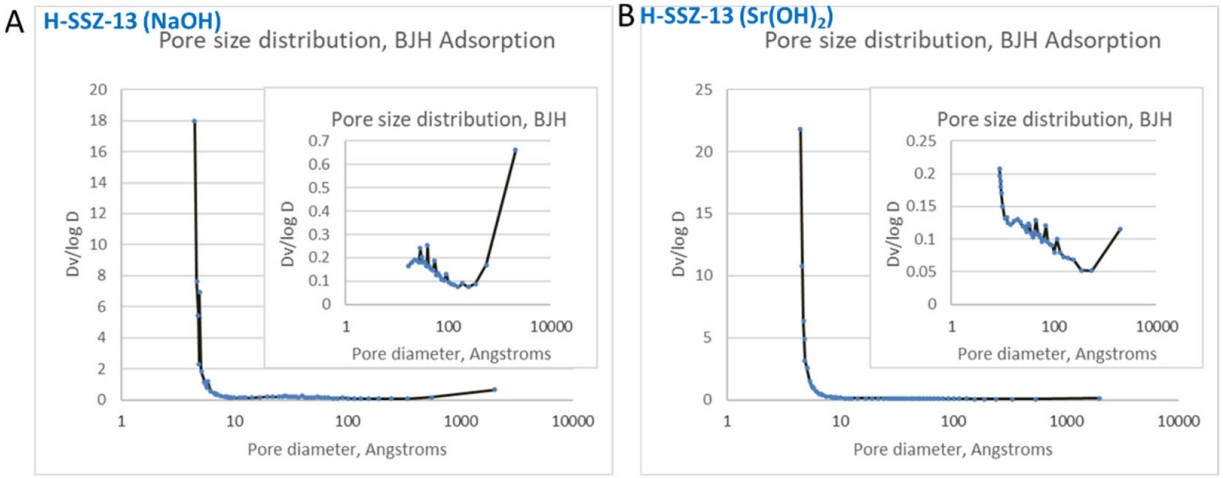

**Figure 6.** BJH pore distribution analysis for nitrogen adsorption curves for both samples.

**Table 1.** Sample Characteristics Assessed by BET Nitrogen Adsorption Measurements.

| Sample ID | Degas Temperature | BET Surface Area | T method Micropore Surface Area | Pore Volume | T method Micropore Surface Area |
|---|---|---|---|---|---|
| | °C | $m^2/g$ | $m^2/g$ | cc/g | cc/g |
| H-SSZ-13 (Na) | 250 | 623 | 409 | 0.85 | 0.16 |
| H-SSZ-13 (Sr) | 250 | 634 | 485 | 0.43 | 0.19 |

BET surface area and pore volume data are summarized in Table 1: both samples have a high surface area of 630 $m^2/g$ (close for both samples) and a high micropore surface area and micropore volume 0.16 and 0.19 cc/g, respectively. These show that the synthesis in the presence of $Sr(OH)_2$ produces a highly crystalline SSZ-13 material with a high surface area and micropore volume similar to the conventional SSZ-13 crystals.

*2.2. Probing Al Pairing with Co(II) Titration, NO Adsorption Measurement and Infra-Red Spectroscopy*

The issue of controlling Al siting in zeolites is a very complex and unresolved one [39–56]. It is universally accepted that the measure of the number of Al pairs in zeolitic materials can be determined by Co(II) ion exchange capacity [38,57–61]. Co(II) ions are generally assumed to exist almost exclusively as Co(II)/2Al species in zeolites. We exchanged both $NH_4$-SSZ-13 samples with Co(II) ions under identical conditions and calcined in the air flow at 650 °C. The resulting blue-powders with isolated Co(II) ions in distorted tetrahedral environment [62] on air slowly (5–7 days) regain the pink color due to moisture adsorption and coordination of moisture to tetrahedral Co to form square-pyramidal and octahedral Co aqua complexes (Figure 3C) [62]. X-ray absorption near edge structure (XANES) spectroscopy analysis of both samples shows that Co is six-coordinate and present in a +2 oxidation state [63,64]. XANES for the Co sample prepared from SSZ-13(Sr) showed better signal intensity than for the one prepared from SSZ-13(Na), already suggesting to us that cobalt is present in higher amounts (Figure 1B and Figure S1) [samples were prepared with the identical masses of sample and thus identical pressed pellets]. With infra-red spectroscopy we studied NO adsorption on Co/SSZ-13 material (Figure 3A). NO adsorption produces two NO stretches that grow in concert upon NO titration. They belong to $Co(II)(NO)_2$ complex [62,65–72] with symmetric NO stretch at 1902 $cm^{-1}$ and asymmetric NO stretch at 1816 $cm^{-1}$. This is typical for atomically-dispersed Co(II) ions in zeolite [62,65–72]. Co ions in a synthesized SSZ-13 sample exist in extra-framework positions and not in silanol nests/silica matrix: the latter vibrate at slightly lower frequencies [62,65–72]. Importantly, 1 cobalt ion can adsorb up to 2 NO molecules.

Co(NO)$_2$ complexes survive under vacuum at elevated temperature (Figure 3A). This suggested to us Co/SSZ-13 materials would be useful for NO adsorption from ambient air. NO is a nuisance pollutant and Co-containing systems could potentially show promise for NO adsorption from ambient air. For these materials, it would be important to maximize the number of Co(II) ions per SSZ-13 mass: the amount of adsorbed cobalt correlates linearly with the number of Al pairs in siliceous SSZ-13 material.

Indeed, both Co/SSZ-13(Na) and Co-SSZ-13(Sr) are active for NO adsorption (Figure 2A). However, the material derived from Sr/SSZ-13 stores at least 55% more NO compared with Co-SSZ-13(Na). Therefore, it is far more advantageous for NO adsorption applications, and consequently it lasts long as a NO adsorber under normal conditions from ambient air (Figure 2B).

The higher number of Al pairs is the direct consequence of Sr(OH)$_2$ used during SSZ-13 synthesis due to the necessity to maintain the Sr-form of zeolite in which Sr(II) cations have a +2 charge and have to be charge balanced by 2 Al atoms residing close to each other, thus allowing for the significantly higher exchange capacity of atomically dispersed divalent cations. This is consistent with EDS mapping results showing that the Co amount in the Co-SSZ-13 (Sr) sample is 3.41 wt% (Figure 7), consistent with the ICP result for Co in that sample, 3.48%. The conventional sample has 2.20 wt% Co based on ICP analysis. This is further in agreement with a >50% increase of NO adsorption capacity of the Co-SSZ-13 (Sr) sample.

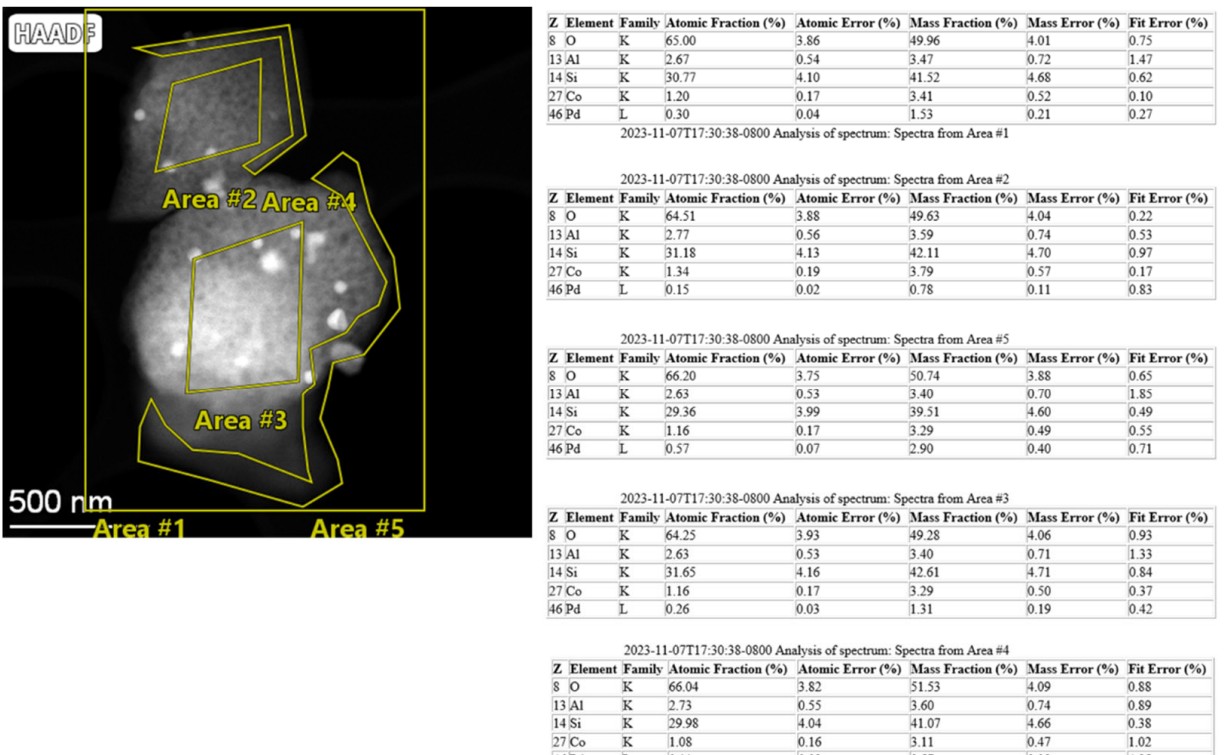

2023-11-07T17:30:38-0800 Analysis of spectrum: Spectra from Area #1

| Z | Element | Family | Atomic Fraction (%) | Atomic Error (%) | Mass Fraction (%) | Mass Error (%) | Fit Error (%) |
|---|---------|--------|---------------------|------------------|-------------------|----------------|---------------|
| 8 | O | K | 65.00 | 3.86 | 49.96 | 4.01 | 0.75 |
| 13 | Al | K | 2.67 | 0.54 | 3.47 | 0.72 | 1.47 |
| 14 | Si | K | 30.77 | 4.10 | 41.52 | 4.68 | 0.62 |
| 27 | Co | K | 1.20 | 0.17 | 3.41 | 0.52 | 0.10 |
| 46 | Pd | L | 0.30 | 0.04 | 1.53 | 0.21 | 0.27 |

2023-11-07T17:30:38-0800 Analysis of spectrum: Spectra from Area #2

| Z | Element | Family | Atomic Fraction (%) | Atomic Error (%) | Mass Fraction (%) | Mass Error (%) | Fit Error (%) |
|---|---------|--------|---------------------|------------------|-------------------|----------------|---------------|
| 8 | O | K | 64.51 | 3.88 | 49.63 | 4.04 | 0.22 |
| 13 | Al | K | 2.77 | 0.56 | 3.59 | 0.74 | 0.53 |
| 14 | Si | K | 31.18 | 4.13 | 42.11 | 4.70 | 0.97 |
| 27 | Co | K | 1.34 | 0.19 | 3.79 | 0.57 | 0.17 |
| 46 | Pd | L | 0.15 | 0.02 | 0.78 | 0.11 | 0.83 |

2023-11-07T17:30:38-0800 Analysis of spectrum: Spectra from Area #5

| Z | Element | Family | Atomic Fraction (%) | Atomic Error (%) | Mass Fraction (%) | Mass Error (%) | Fit Error (%) |
|---|---------|--------|---------------------|------------------|-------------------|----------------|---------------|
| 8 | O | K | 66.20 | 3.75 | 50.74 | 3.88 | 0.65 |
| 13 | Al | K | 2.63 | 0.53 | 3.40 | 0.70 | 1.85 |
| 14 | Si | K | 29.36 | 3.99 | 39.51 | 4.60 | 0.49 |
| 27 | Co | K | 1.16 | 0.17 | 3.29 | 0.49 | 0.55 |
| 46 | Pd | L | 0.57 | 0.07 | 2.90 | 0.40 | 0.71 |

2023-11-07T17:30:38-0800 Analysis of spectrum: Spectra from Area #3

| Z | Element | Family | Atomic Fraction (%) | Atomic Error (%) | Mass Fraction (%) | Mass Error (%) | Fit Error (%) |
|---|---------|--------|---------------------|------------------|-------------------|----------------|---------------|
| 8 | O | K | 64.25 | 3.93 | 49.28 | 4.06 | 0.93 |
| 13 | Al | K | 2.63 | 0.53 | 3.40 | 0.71 | 1.33 |
| 14 | Si | K | 31.65 | 4.16 | 42.61 | 4.71 | 0.84 |
| 27 | Co | K | 1.16 | 0.17 | 3.29 | 0.50 | 0.37 |
| 46 | Pd | L | 0.26 | 0.03 | 1.31 | 0.19 | 0.42 |

2023-11-07T17:30:38-0800 Analysis of spectrum: Spectra from Area #4

| Z | Element | Family | Atomic Fraction (%) | Atomic Error (%) | Mass Fraction (%) | Mass Error (%) | Fit Error (%) |
|---|---------|--------|---------------------|------------------|-------------------|----------------|---------------|
| 8 | O | K | 66.04 | 3.82 | 51.53 | 4.09 | 0.88 |
| 13 | Al | K | 2.73 | 0.55 | 3.60 | 0.74 | 0.89 |
| 14 | Si | K | 29.98 | 4.04 | 41.07 | 4.66 | 0.38 |
| 27 | Co | K | 1.08 | 0.16 | 3.11 | 0.47 | 1.02 |
| 46 | Pd | L | 0.11 | 0.02 | 0.57 | 0.08 | 4.85 |

**Figure 7.** EDS maps of the selected parts shown in the HAADF-STEM image. Note: Area 1 refers to the total area of the sample and shows its average composition.

We also tested this material as a PNA material for vehicle exhaust. Despite its ability to absorb NOx during vehicle cold start, Co/SSZ-13 is not viable for NO$_x$ adsorber applications because it releases NO$_x$ too early (Figure 2D). However, it is important to note that the presence of Co ions is not detrimental for NO storage (unlike for Na(I)-containing materials, for example as outlined in our previous study [29]).

### 2.3. Investigation of PNA Performance and Hydrothermal Stability of Co-SSZ-13 Materials

The presence of a higher fraction of paired Al sites in Co/SSZ-13(Sr) compared with Co-SSZ-13(Na) prompted us to investigate whether this sample possesses higher hydrothermal stability. Remarkably, hydrothermal aging of Co-SSZ-13(Sr) at 850 °C and then further 900 °C (for the second time) completely preserves its NO storage capacity/uptake (Figure 2C). Since the Co cations are the active sites for NO adsorption (NO does not adsorb on zeolite in significant amounts without Co), this clearly proves that hydrothermal aging of this material at 900 °C does not lead to cobalt (II) ion removal from the framework or collapse of the SSZ-13 zeolitic structure. This is nicely corroborated by the XRD data in Figure 1, which shows highly crystalline XRD patterns of freshly prepared and hydrothermally aged Co-SSZ-13 (Sr) samples without any evidence of crystallinity loss and CoOx/Co metal formation. This, to the best of our knowledge, is the first demonstration of metal-containing SSZ-13 catalytic material's hydrothermal stability at temperatures as high as 900 °C. This further corroborates our initial suggestion that distribution of Al sites within a zeolite framework is a key factor in zeolitic hydrothermal stability: the material with a greater number of Al pairs (synthesized in the presence of alkaline earth metal hydroxide in the synthesis gel) presents notably higher hydrothermal stability than its SSZ-13(Na) analogue when exchanged with divalent metals (Figure S2).

### 2.4. Investigation of Hydrothermal Stability and PNA Performance of Pd-Containing SSZ-13 Materials

This remarkable property of Co/SSZ-13(Sr) material, thus, prompted us to load 1.7 wt% Pd in its micropores and study its PNA properties and the hydrothermal stability of the resulting Pd-containing SSZ-13 composite [20,21,23–38]. Indeed, the sample shows excellent PNA performance after loading Pd in it (Figure 8A).

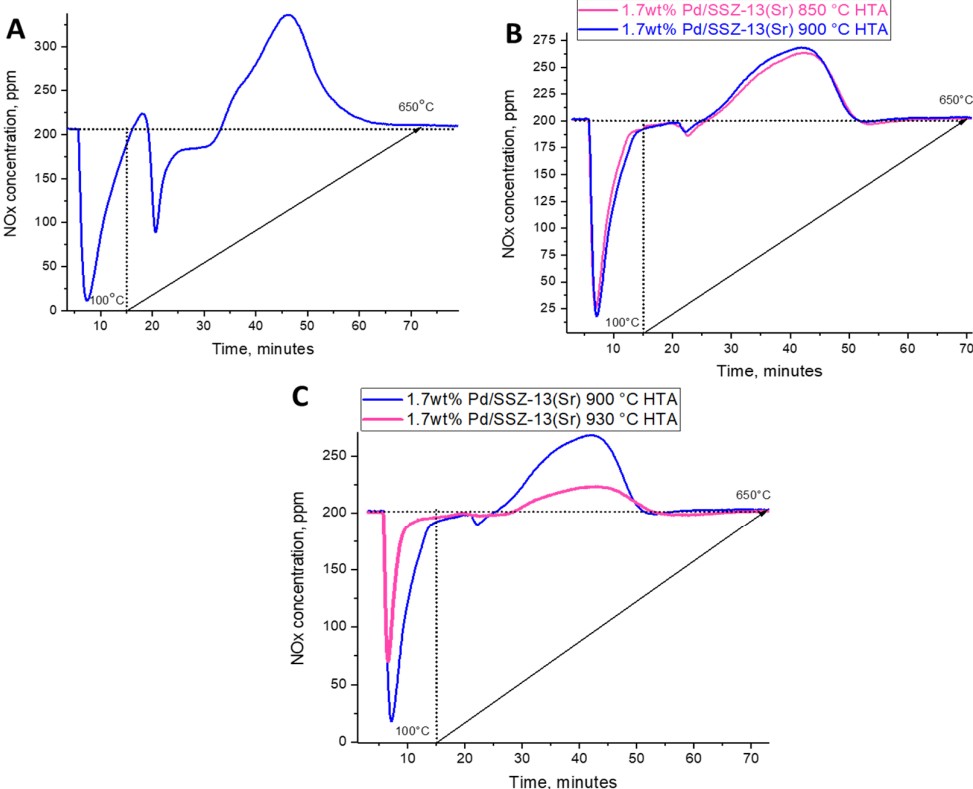

**Figure 8.** (**A**) NOx adsorption on "fresh" 1.7 wt% Pd/Co-SSZ-13 sample derived from an Sr(OH)$_2$ synthetic route at 100 °C for 10 min (after 5 min bypass) followed with TPD (10 °C/min). For simplicity the sample is denoted as Pd-SSZ-13 (Sr). The feed gas mixture contains 220 ppm of NOx,

250 ppm CO, 14% $O_2$ and 3% $H_2O$. Sample mass 120 mg. Total flow rate 300 sscm/min. (**B**) NOx adsorption on 850 and 900 °C hydrothermally aged 1.7 wt% Pd/Co-SSZ-13 sample derived from an Sr(OH)$_2$ synthetic route at 100 °C for 10 min (after 5 min bypass) followed with TPD (10 °C/min). The feed gas mixture contains 220 ppm of NOx, 250 ppm CO, 14% $O_2$ and 3% $H_2O$. Sample mass 120 mg. Total flow rate 300 sscm/min. (**C**) NOx adsorption on 900 and 930 °C hydrothermally aged 1.7 wt% Pd/Co-SSZ-13 sample derived from an Sr(OH)$_2$ synthetic route at 100 °C for 10 min (after 5 min bypass) followed with TPD (10 °C/min). The feed gas mixture contains 220 ppm of NOx, 250 ppm CO, 14% $O_2$ and 3% $H_2O$. Sample mass 120 mg. Total flow rate 300 sscm/min.

The second NOx adsorption peak occurs during PNA on fresh Pd-Co-SSZ-13(Sr) sample at higher temperatures (that adsorption peak is absent for just Pd-SSZ-13 materials [24–27,29,31,32]). We tentatively attribute it to the presence of Co in the "fresh" sample and possible Co-Pd synergy in the fresh sample. However, we must note that the industry performance of fresh samples is not relevant and the evaluation of hydrothermally aged (de-greened) samples is the standard. Previously, the best known Pd-zeolitic PNA materials could survive prolonged hydrothermal aging of 800–820 °C. However, at higher temperatures they lost most of their performance (see, for example, Pd-SSZ-39 and Pd/FER studies [31,37]). We first hydrothermally aged the novel 1.7 wt% Pd-Co-SSZ-13 material at 850 °C for 10 h in air/steam flow. Surprisingly, the low-temperature NOx adsorption band associated with Pd was fully preserved, and the sample was active as a PNA material (Figure 8B). Encouraged by this unprecedented result (no known SSZ-13 material has been previously demonstrated to have such high hydrothermal stability), we further aged this material (after it was already aged at 850 °C) for the same duration in air/steam flow at 900 °C. To our delight, its PNA performance did not deteriorate (Figure 8B). NO/Pd ratio of the hydrothermally aged material is ~0.8, consistent with utilization of nearly every atomically dispersed Pd cation for NO storage and comparable with the state-of-the-art formulations in the field. This is the most hydrothermally stable known PNA material ever reported. When we pushed it even further and hydrothermally aged this material (for the third consecutive time after it was aged first at 850 °C and then at 900 °C) at 930 °C for 10 h in air/steam flow, ~55% of PNA capacity was lost (the NO/Pd ratio 0.43). However, substantial PNA capacity was preserved, and even such extremely harsh hydrothermal treatment did not lead to full performance loss (Figure 8C).

We further loaded 1.7 wt% Pd on the Co-SSZ-13 (NaOH) sample in order to understand whether that sample would survive hydrothermal aging. This sample was hydrothermally aged at 850 °C and lost most of its NO adsorption capacity (Figure S3), further corroborating the outstanding hydrothermal stability of the Pd supported on Sr-derived catalysts.

XRD data for Pd/Co-SSZ-13 (Sr) sample aged at 930 °C (Figure 1) reveals preservation of the crystallinity of SSZ-13. Therefore, the crystalline phase collapse is not the reason behind the performance loss. It is notable that the SSZ-13 sample derived via the Sr route survives hydrothermal aging at 930 °C. We previously showed that dealumination leads to removals of Pd from the pores and formation of Pd/PdO particles on the external surfaces of the zeolite. Careful evaluation of the XRD pattern of this sample demonstrates (Figure S4) the presence of Pd metal and Pd oxide in this sample. Thus, 930 °C hydrothermal aging leads to the elution of Pd ions from the framework with the formation of Pd metal and PdO on the external surface of the zeolite (the loss of atomically dispersed Pd sites, which are the binding sites for NO, is the reason behind it).

To test this hypothesis, we turned to HAADF-STEM microscopy on the Pd/Co-SSZ-13 sample hydrothermally aged at 930 °C (Figure 9).

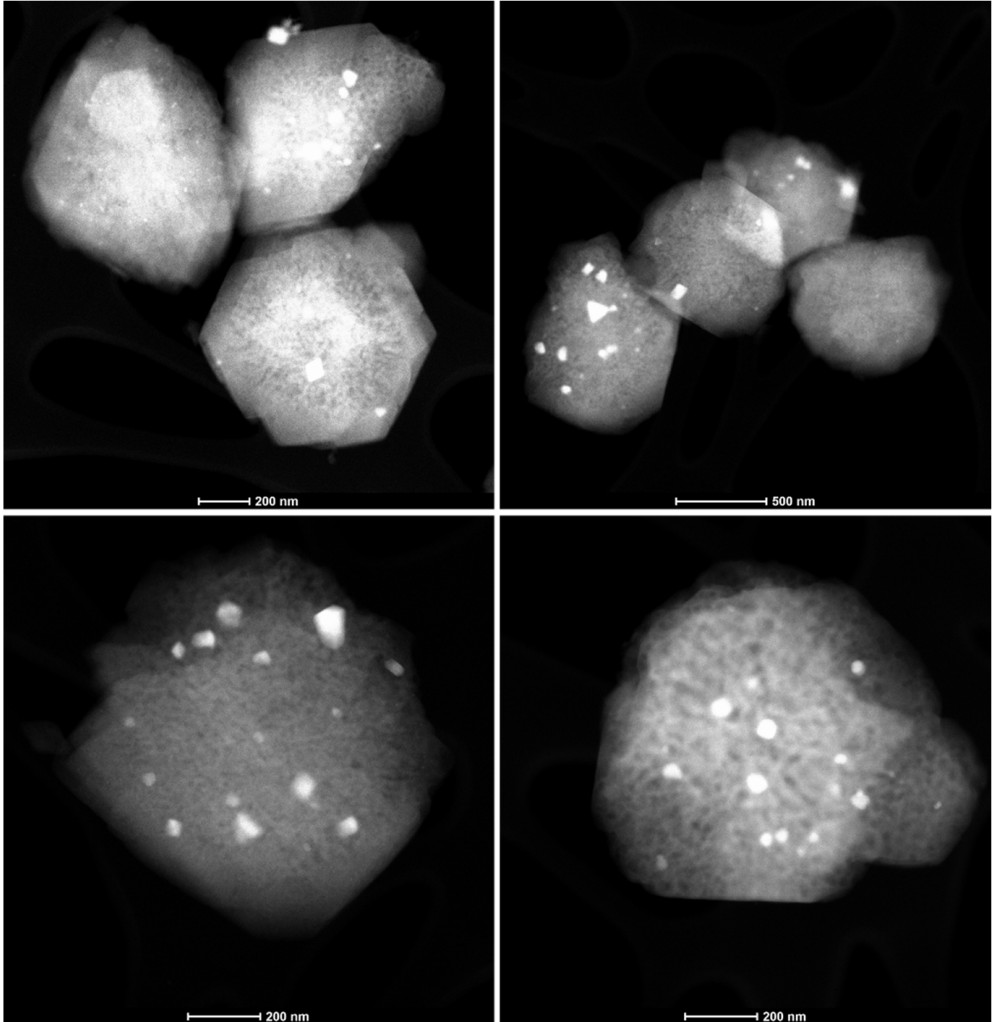

**Figure 9.** Representative HAADF-STEM images of a 930 °C hydrothermally aged 1.7 wt% Pd/Co-SSZ-13 catalyst. The bright spots decorating the surfaces are Pd and PdO nanoparticles. It is obvious that the crystals are zeolite crystals that are preserved; however, formation of mesopores inside the crystals after steaming is obvious (darker areas inside the zeolite crystals).

Interestingly, the crystals retain their shapes consistent with preservation of crystallinity. However, Pd is eluted from micropores during such harsh aging and forms rather large Pd and PdO nanoparticles on the external surface of zeolite crystals, consistent with significant loss of adsorption capacity after aging at 930 °C. The formation of mesopores inside the zeolite crystals is obvious from Figure 10 as well. This is the consequence of dealumination and Al/Si site redistribution during hydrothermal aging. The fate of these eluted Al species becomes cleared in the bright-field HRTEM images (Figure 10). Some of them form alumina clusters on the surfaces of the crystal while a part forms alumina agglomerates outside the crystals. To the best of our knowledge, this is the first visualization of the fate of an eluted framework Al species from zeolite.

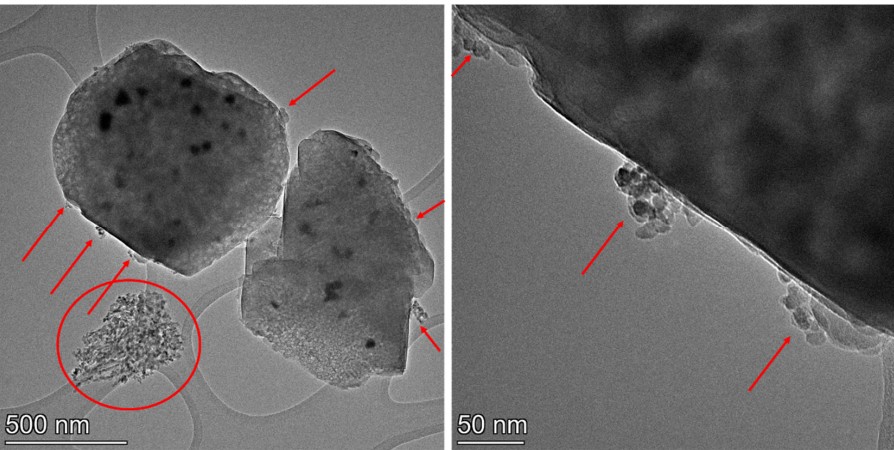

**Figure 10.** HRTEM images of a 930 °C hydrothermally aged 1.7 wt% Pd/Co-SSZ-13 catalyst. Red circle shows agglomeration of alumina clusters. Red arrows show alumina clusters formed on the external surfaces of zeolite crystals.

We also performed EDS mapping on selected crystals of aged Pd/Co-SSZ-13 (Figures 7, 11 and 12).

From these maps, elution of Pd/PdO nanoparticles from the extraframework positions of SSZ-13 with the formation of rather large Pd and PdO nanoparticles becomes obvious. This leads to the loss of NO storage sites. However, it is obvious that the regions devoid of Pd particles in the maps still contain ~0.5–1 wt% of atomically dispersed Pd; thus, the NOx storage for materials after such harsh exposure is retained at the ~50% level. The loss of Pd is caused by the elution of Al from the framework position during hydrothermal aging, which, in turn, leads to expulsion of Pd from the micropores.

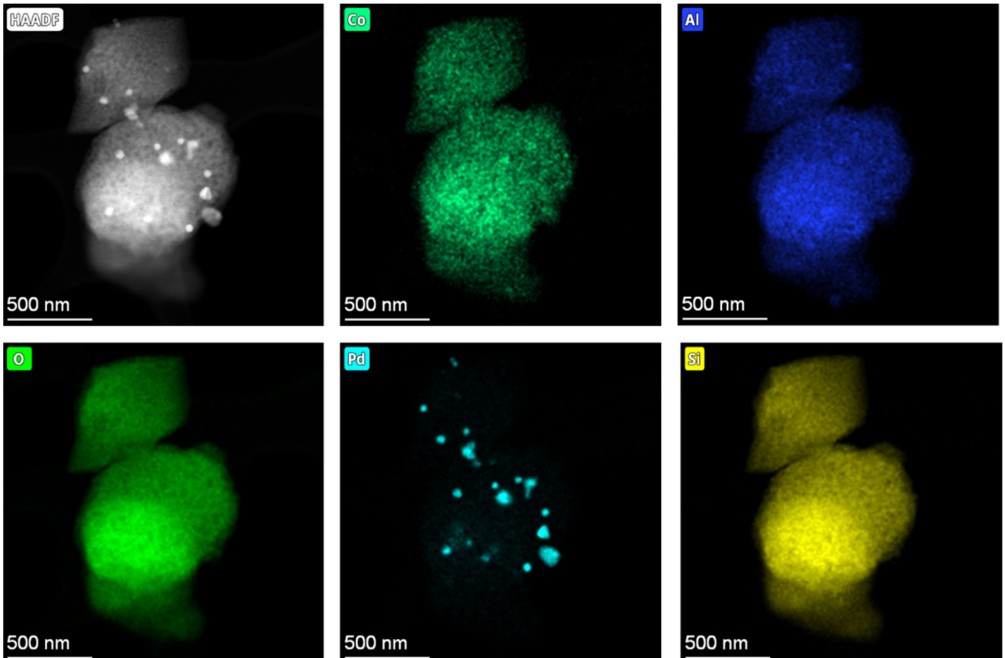

**Figure 11.** EDS maps of selected area of a 930 °C hydrothermally aged 1.7 wt% Pd/Co-SSZ-13 catalyst.

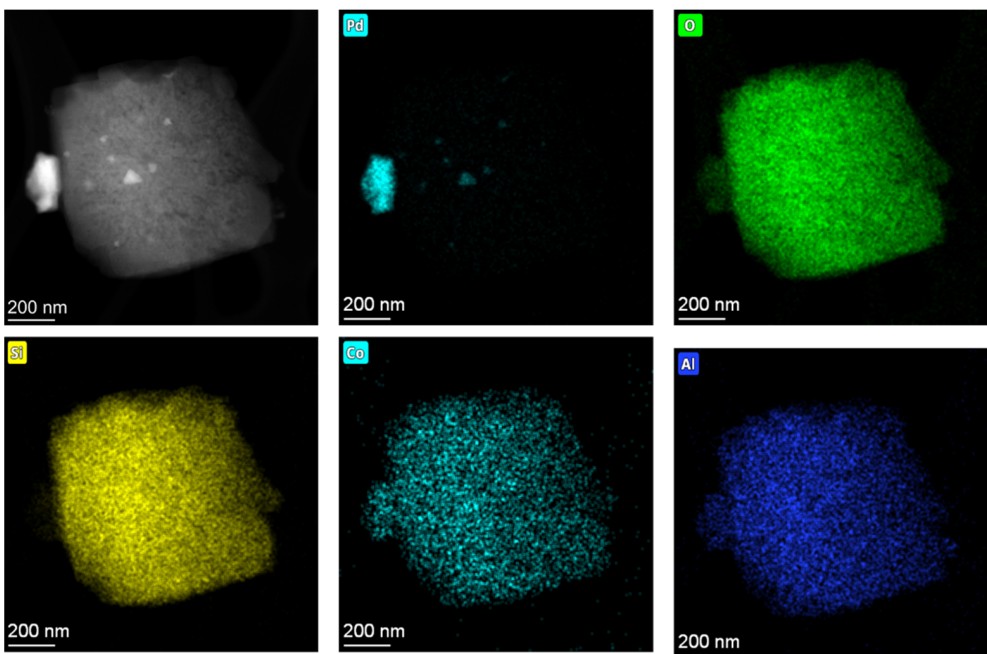

**Figure 12.** EDS maps of another selected area of a 930 °C hydrothermally aged 1.7 wt% Pd/Co-SSZ-13 catalyst.

*2.5. Synthesis and Stability Evaluation of Cu/SSZ-13 Catalysts for SCR of Nitric Oxide*

So far, we have been able to demonstrate the utility of the newly discovered synthesis of SSZ-13 for the preparation of improved M(II)/SSZ-13 materials that can be effectively utilized for adsorption applications. To show the generality of this new siliceous SSZ-13 (Sr) with increased fraction of Al pairs, we extended our study additionally to catalytic applications of the SSZ-13 (Sr). The most relevant example of catalytic use of SSZ-13 is its Cu(II)-exchanged form, industrially deployed for selective catalytic reduction (SCR) of nitric oxide with ammonia [4–7,12–20]. Conventional Cu/SSZ-13 materials are very effective for SCR, yet in diesel engines they are periodically, continuously exposed to temperatures on the order of 800–850 °C, especially in so-called close-coupled engine configurations where the engine exhaust passing through the catalyst is much hotter. Furthermore, exposure to extremely harsh temperatures (often reaching in excess of 900 °C) for relatively short periods also occurs. So far, no known Cu/SSZ-13 material was shown to withstand hydrothermal aging at 850 °C and above without significant catalytic activity deterioration [64,73–79]. Indeed, >80% loss of activity is observed for typical Cu-SSZ-13 as well established in the literature [64,73]. In order to solve the existing issue with the hydrothermal stability of Cu/SSZ-13 catalysts, we prepared a Cu/SSZ-13 (Sr) catalyst and subjected it to extremely harsh hydrothermal aging at 920 °C. In excellent agreement with the high hydrothermal stability of Co- and Pd-SSZ-13 materials, Cu/SSZ-13(Sr) showed excellent performance for NO reduction even after exposure to such extreme hydrothermal aging conditions (Figure S5). This highlights the advantage of the novel synthesis method for siliceous zeolites with increased number of Al pairs that we report in the present study in the preparation of stable materials. The performance of the aged Cu-SSZ-13 materials prepared through the regular route and under conditions similar or identical to ours shows great loss in all the studied temperature ranges [64,73–79].

**3. Experimental Methods**

Synthesis of Na/SSZ-13 zeolite with Si/Al~10 was prepared according to method B described in [1] from the gel of composition:

$$100 \ SiO_2 : 4.8 \ Al_2O_3 : 20 \ NaOH : 20 \ TMAda : 4400 \ H_2O.$$

First, 1.6 g of NaOH (Sigma-Aldrich, Saint Louis, MO, USA) was dissolved in 132 g of DI water. Next, 1.5 g of Al(OH)$_3$ was added followed by the addition of 34.1 g of TMAda-OH (N,N,N-trimethyl-adamantyl ammonium hydroxide solution 24.8 wt.% in water, Sachem Zeogen 2585). The solution was stirred until it become clear. Finally, 12 g of fumed silica (0.007 μm average particle size, Sigma-Aldrich) was slowly added under vigorous stirring. The prepared gel was aged at room temperature for 24 h (stirring conditions) and subsequently placed in a Teflon-lined autoclave. The hydrothermal synthesis was carried out in rotated autoclaves (60 rpm) at 160 °C for 96 h. The solid product was separated from the mother liquid via centrifugation and washed several times until a neutral pH could be measured. The synthesized zeolite was then dried in an oven at 100 °C overnight. Calcination, necessary to decompose the organic template, was carried out in a tube furnace at 550 °C in flowing synthetic air (100 mL/min) for 8 h (heating rate 1 °C/min).

Synthesis of SSZ-13 with Si/Al~10 in the presence of Sr(OH)$_2$ was carried out using a similar recipe as for the Na/SSZ-13; however, in this case, the mole-equivalent amount of strontium hydroxide (Sigma-Aldrich) was used instead of NaOH. The gel composition was:

$$100 \text{ SiO}_2 : 4.8 \text{ Al}_2\text{O}_3 : 10 \text{ Sr(OH)}_2 : 35.2 \text{ TMAda} : 4700 \text{ H}_2\text{O}.$$

Preparation of the gel involved adding 5.3 g of Sr(OH)$_2$ to 108.8 g of DI H$_2$O followed by the addition of 1.5 g of Al(OH)$_3$. Subsequently, 60.1 g of TMAda was added, and obtained suspension containing not completely dissolved strontium hydroxide was stirred at room temperature for 21 h. Then, 12 g of fumed silica was slowly added, and the prepared gel was aged for 24 h at ambient temperature and under stirring before loading in Teflon-lined autoclaves. The synthesis was carried out in rotated (60 rpm) autoclaves at 160 °C and kept under these conditions for 4 days. The synthesized microporous product was separated from the suspension by centrifugation and washed multiple times with deionized water.

Drying of the obtained Sr/SSZ-13 zeolite, as well as subsequent calcination, was done in the same conditions as those used for Na/SSZ-13. Cobalt(II) ion-exchange was performed by dispersing 1 g of NH$_4$-SSZ-13(Na) and NH$_4$-SSZ-13(Sr) powders in excess (150 mL) 2 M cobalt(II) nitrate (Sigma-Aldrich) solution at 80 °C with continuous stirring for 8 h. Then, the powder was separated by centrifugation and washed with excess DI water 3 times (with centrifugation after each wash), dried at 80 °C in air flow and subsequently calcined at 650 °C for 4 h.

Palladium tetramine nitrate (10% solution in water, Sigma-Aldrich) was used to load 1.7 wt% Pd in Co-SSZ-13. More specifically, the desired amount of palladium precursor solution was mixed with a minimum amount of water (the total volume approximately equivalent to the pore volume of zeolite ~0.25 mL/g), then added drop-wise with a micropipette to Co-SSZ-13 powder.

Copper nitrate hydrate (>99.999% Sigma-Aldrich) was used to load 2.5 wt% copper in SSZ-13. More specifically, the desired amount of copper precursor solution was mixed with a minimum amount of water (the total volume approximately equivalent to the pore volume of zeolite ~0.25 mL/g), then added drop-wise with a micropipette to NH$_4$-SSZ-13 powder. The resulting powder was dried at 80 °C in the air flow and subsequently calcined at 650 °C for 4 h.

Standard NOx adsorption tests were conducted in a plug-flow reactor system with powder samples (120 mg, 60–80 mesh) loaded in a quartz tube, using a synthetic gas mixture that contained ~200–220 or 27 ppm of NO, 250 ppm or 0 CO, 300 ppm or 3% H$_2$O, 14% or 20% O$_2$) balanced with N$_2$ at a flow rate of 300 sscm/min.

Standard NH3-SCR (4NO + 4NH$_3$ + O$_2$ = 4N$_2$ + 6H$_2$O) "light-off" tests were conducted on a custom-built plug-flow SCR reaction test stand equipped with an online MKS MultiGas 2030 FTIR gas analyzer (Suite 201 Andover, MA, USA) with the gas cell retained at 191 °C for measuring concentrations of the reactants and products. A mass of 120 mg of sieved

catalyst (60–80 mesh) was supported on a quartz frit inside a 0.8-cm inner diameter quartz reactor placed in an electric tube furnace. The composition of the gas feed was 360 ppm $NH_3$, 360 ppm NOx, 14% $O_2$, 2.4% $H_2O$ and balance $N_2$. The total flow rate was 300 mL/min.

All the gas lines were heated to over 100 °C. Concentrations of reactants and products were measured by an online MKS MultiGas 2030 FTIR gas analyzer with a gas cell maintained at 191 °C. Two four-way valves were used for gas switching between the reactor and the bypass. Prior to storage testing at 100 °C, the sample was pretreated in air flow for 1 h at 550 °C and cooled to the target temperature in the same feed. The gas mixture was then switched from the reactor to the bypass, and the desired concentration of NOx was added to the mixture. Upon stabilization, the gas mixture was switched back from bypass to the reactor for storage testing for 10 min. The sample was then heated to 650 °C at a rate of 10 °C/min to record the desorption profiles of gases in the effluent. Ambient-temperature measurements were performed with 120 mg of sample at 23 °C by flowing ~27 ppm NO (with 20% oxygen, 300 ppm water balanced in nitrogen) at 300 sscm/min rate at constant temperature.

Hydrothermal aging was performed in a plug-flow reactor in 10% $H_2O$/Air at flow at GHSV 150 L/g*h. Standard conditions include 10 h hydrothermal aging unless noted otherwise. The temperature of aging for each material is provided in the text (between 800 and 930 °C).

XANES data collection on Co/SSZ-13 samples was performed on a bench-top easyXAFS300 instrument (easyXAFS, Renton, WA, USA). Details about the instrument can be found elsewhere [80]. One hundred twenty mg of each Co-SSZ-13 sample was pressed into a pellet (13 mm diameter) in order to obtain a sufficient count rate for the quite low amount of Co in the zeolite matrix. The pellet was wrapped into a polyimide (Kapton) tape to maintain its integrity. Co K-edge spectra were collected using a Ge444 spherically bent crystal analyzer and Mo anode X-ray tube. The data quality was assured by comparing the Co K-edge scan of Co foil collected with the easyXAFS300 instrument to the Co K-edge scan of Co foil collected at the synchrotron facility and available in the Hephaestus standards database, Demeter version 0.9.26, Figure 13. Spectra were deadtime corrected and the energy edge was calibrated using the Co foil standard, accounting for the non-linear shift in energy. The Co-SSZ-13(Sr) sample had a higher amount of Co, which allowed us to collect high-quality XANES data with an edge step of 0.54; the sample Co-SSZ-13(Na) did not yield in the good quality spectra due to a lower amount of Co in the zeolite matrix; the edge step was 0.3 and the white line was suppressed due to the sorbing matrix of zeolite, Figure 1.

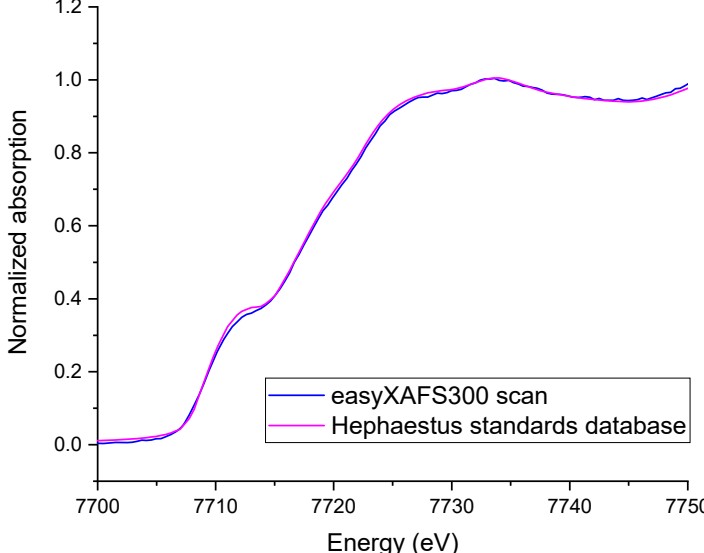

**Figure 13.** Co K-edge XANES spectra collected for Co foil with the lab scale instrument (easyXAFS300) and at the synchrotron facility (NSLS X11B, Hephaestus standards database, Demeter version 0.9.26 B.

The in situ static transmission IR experiments were conducted in a home-built cell housed in the sample compartment of a Bruker Vertex 80 spectrometer, equipped with an MCT detector and operated at 4 cm$^{-1}$ resolution. The powder sample was pressed onto a tungsten mesh, which, in turn, was mounted onto a copper heating assembly attached to a ceramic feedthrough. The sample could be resistively heated, and the sample temperature was monitored by a thermocouple spot welded onto the top center of the W grid. The sample was evacuated at 200 °C prior to measurement to remove moisture. NO was cleaned with multiple freeze–pump–thaw cycles. Prior to spectrum collection, a background with the activated (annealed at 300 °C) sample in the IR beam was collected. Each spectrum reported is obtained by averaging 256 scans.

### 3.1. Surface Area and Pore Structure Characterization

Surface area and pore structure analysis were measured by $N_2$ adsorption at 77 K with an automatic gas sorption system: Autosorb iQ Gas Sorption System from Quantachrome Instruments (a brand of Anton Paar; Ashland, VA, USA). The samples were degassed under vacuum at 250 °C for 12 h before the adsorption measurements. The surface area was determined using the 7 points BET (Brunauer-Emmett-Teller) method (P/P$_0$: $5 \times 10^{-3}$ P/P$_0$—$2 \times 10^{-2}$). The BJH (Barrett, Joyner & Halenda) method and the Non-Local Density Functional Theory (NLDFT) method were used for the pore size distribution determination.

### 3.2. Microscopy

Structural analysis was performed with aberration corrected Thermo-Fisher Themis Z Scanning/Transmission Electron Microscope (S/TEM) (Waltham, MA, USA). The observations were performed in Scanning mode using a HAADF detector. The probe convergence angle was 25 mrad, and the inner detection angle on the HAADF detector was 52 mrad.

### 3.3. X-ray Diffraction (XRD) Measurements

X-ray diffraction data were collected using a Rigaku SmartLab XE diffractometer (Tokyo, Japan) equipped with a Cu rotating anode (λ = 1.5418 Å) operated at 45 kV, 200 mA.

### 4. Conclusions

In summary, we report a simple and scalable method of increasing Al pairs distribution in siliceous SSZ-13 via synthesis in the presence of Sr(OH)$_2$. We show that Al distribution is key in exchanging higher amounts of divalent metal cations (Co(II) and Pd(II), for example) atomically dispersed in the SSZ-13 micropores. The resulting materials show the ability to adsorb NOx pollutants under various conditions. Furthermore, we show remarkable hydrothermal stability of the resulting metal zeolite materials that survive extraordinarily harsh hydrothermal aging conditions (encountered in vehicle exhaust under certain circumstances) at 900 °C and that do not lose full NOx adsorption capacity at even higher temperatures (930 °C) for both catalytic and adsorption applications. Furthermore, Cu-SSZ-13 catalysts with unprecedented hydrothermal stability (up to 920 °C) were prepared on the basis of SSZ-13 with a higher fraction of Al pairs (Sr-derived). Our study unveils a hitherto unexplored pathway to prepare zeolites and metal/zeolite composites with tailored fractions of paired sites and advantageous properties (stability, activity) not available before.

**Supplementary Materials:** The following supporting information can be downloaded at: https://www.mdpi.com/article/10.3390/catal14010056/s1, Figure S1: EDS mapping chosen on different areas of H-SSZ-13 (Sr) crystals. The corresponding tabulated values are shown in Table S1. Figure S2: NOx adsorption on Co-SSZ-13 sample (derived from NaOH and hydrothermally aged at 870 °C) at 100 °C for 20 min (after 10 min bypass). The feed gas mixture contains 220 ppm of NOx, 14% $O_2$ and 300 ppm $H_2O$. Sample mass 120 mg. Total flow rate 300 sscm/min. It is obvious that only a relatively small uptake of NO is observed, consistent with much lower hydrothermal stability of Co-SSZ-13 (Na) sample. If the structure were preserved, no loss of activity would be observed. Figure S3:

NOx adsorption on 850 °C hydrothermally aged 1.7 wt% Pd/Co-SSZ-13 sample derived from NaOH synthetic route at 100 °C for 5 min (after 10 min bypass). The feed gas mixture contains 220 ppm of NOx, 250 ppm CO, 14% $O_2$ and 3% $H_2O$. Sample mass 120 mg. Total flow rate 300 sscm/min. Rather small uptake of NO occurs, consistent with material degradation. Figure S4: XRD pattern of Pd/SSZ-13 (Sr) sample hydrothermally aged at 930 °C. Blue ticks indicate the SSZ-13 crystalline phase reflections. Red ticks indicate the Pd metal presence. Green ticks indicate PdO presence. Figure S5: NO conversion vs Temperature. Conditions: 120 mg, 2.5 % wt% Cu/SSZ-13(Sr), 360 ppm $NH_3$, 360 ppm NO, 14% $O_2$, 2.4 % $H_2O$ and balanced with $N_2$. The total flow rate 300 ml/min. GHSV 150 L/g*hr. The sample was aged at 920 °C in air/10% $H_2O$ flow for 3 h before catalysis. Table S1: Atomic and weight percentages of Si, Al, and O EDS mapped in Figure 1.

**Author Contributions:** Conceptualization, K.K. and M.A.D.; methodology, K.K. and M.A.D.; validation, K.K., M.A.D. and J.S.; formal analysis, K.K. and M.A.D.; investigation, K.K. and M.A.D.; resources, K.K., M.A.D., J.S., C.P. and Y.W.; data curation, K.K., M.A.D., L.K., M.B., X.S.L., N.R.J., D.B., X.I.P.-H., C.P., Y.W. and J.S.; writing—original draft preparation, K.K. and M.A.D.; writing—review and editing, K.K.; visualization, K.K.; supervision, K.K. and M.A.D.; project administration, K.K. and M.A.D.; funding acquisition, K.K., M.A.D., J.S., C.P. and Y.W. All authors have read and agreed to the published version of the manuscript.

**Funding:** We would like to thank the U.S. Department of Energy, Energy Efficiency and Renewable Energy, Vehicle Technology Office for the support. We would like to thank the financial support by Crosscut Lean Exhaust Emissions Reduction Simulations (CLEERS), which is an initiative funded by the U.S. Department of Energy (DOE) Vehicle Technologies Office to support the development of accurate tools for use in the design, calibration and control of next-generation engine/emissions control systems that maximize efficiency while complying with emissions regulations. MAD was supported by the Materials Synthesis and Simulation Across Scales (MS3) Initiative conducted under the Laboratory Directed Research & Development Program at PNNL. D.B. and C.P. were supported by the Physical & Computational Sciences Directorate Science Mission Seed LDRD Program at PNNL. Most experiments were conducted in the Environmental Molecular Sciences Laboratory (EMSL), a national scientific user facility sponsored by the Department of Energy's Office of Biological and Environmental Research at Pacific Northwest National Laboratory (PNNL). PNNL is a multi-program national laboratory operated for the DOE by Battelle.

**Data Availability Statement:** The data in this article available in the article and Supplementary Materials.

**Conflicts of Interest:** The authors declare no conflicts of interest.

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
