# Peer review of "Increasing Al-Pair Abundance in SSZ-13 Zeolite via Zeolite Synthesis in the Presence of Alkaline Earth Metal Hydroxide Produces Hydrothermally Stable Co-, Cu- and Pd-SSZ-13 Materials"

_catalysts, doi:10.3390/catal14010056_

Round 1

Reviewer 1 Report

Comments and Suggestions for Authors

1.        In this paper, alkaline earth metal doped synthetic gel was used to induce SSZ-13 zeolite with more Al pairs. What is the reason for choosing Sr compared with Ca, Mg, etc., which are more common? Whether a comparative experiment can be added?

2.        It is confirmed in Sec. 2.1 that the Co(II) ion-exchanged into SSZ-13 prepared from the Sr contained system atomically dispersed Co(II) ions in zeolite, thus presenting good thermal stability. What about the Pd species and Cu species variation in Sr-SSZ-13 compared with Na-SSZ-13? Some characterization is necessary.

3.        It is suggested the detailed NOx adsorption capacity be calculated for both Co-SSZ-13 and Pd-SSZ-13 in this work, which should be compared with other researchers’ report for clearly show the superiority of this synthesis method.

4.        In this paper, only experimental data were obtained when Sr-SSZ-13 supported Pd and Cu adsorbed or catalyzed NOx. It is suggested that the performance test data of Pd/SSZ-13 and Cu/SSZ-13 materials without Sr should be added to form a control experiment, which is more convincing.

5.        In this paper, the hydrothermal aging properties of M/SSZ-13(Sr) material at 900℃ have been investigated and good experimental results have been obtained. Does the structure of SSZ-13 change at 900℃? If the change occurs, will it affect the NOx adsorption/catalytic performance of the material?

Reviewer 2 Report

Comments and Suggestions for Authors

Comments to the paper “Increasing Al-pair abundance in SSZ-13 zeolite via zeolite synthesis in the presence of alkaline earth metal hydroxide produces hydrothermally stable Co-, Cu- and Pd-SSZ-13 materials” by Konstantin Khivantsev, Miroslaw A. Derewinski, Nicholas R. Jaegers, Daria Boglaienko, Xavier I. Pereira-Hernandez, Carolyn Pearce, Yong Wang and János Szanyi

A brief summary

The above paper claims the successful synthesis of pure phase, high silica SSZ-13 zeolite from a synthesis gel containing Sr(OH)2 instead of NaOH. The prepared material (SSZ-13(Sr)) was ion-exchanged with Co(NO3)2 solution. The Co containing material showed significantly higher NO adsorption capacity than the Co-zeolite prepared from a NaOH-containing synthesis gel. The paper clearly presents that the NOx adsorption capacity of the Co-SSZ-13(Sr) and Pd,Co-SSZ-13(Sr) samples was preserved after ageing at very high temperatures. The excellent catalytic activity of the Cu-SSZ-13(Sr) sample after ageing at 920 °C is also presented.

General comments

-          The names of the zeolite samples throughout the manuscript are not unified.

-          The XRPD patterns and elemental analysis (AAS or ICP) of Co and Al content of the prepared samples are missing. Co/Al ratio should be given. Advantageously Co/AlF ratio as well. (AlF: framework Aluminum)

-          Section 2.1 is about probing Al pair sites in the framework. Is the presence of Co(NO)2 complexes an evidence for “bare” Co(II) ions at Al pair sites?

-          The title of section 2.2 was better “Investigation of PNA performance and hydrothermal stability of Co-SSZ-13 materials” like the title of section 2.3.

-          In section 2.2 it should be discussed in more detail why were different composition for the feed gas mixture chosen in different experiments shown in Fig.2.

-           In section 2.3, the second NOx adsorption peak (at about 20-30 min) in Fig. 3.A would need more discussion. “We, thus, attribute it to the presence of Co in the “fresh” sample.” But, in Fig. 2.D for Co-SSZ-13(Sr), this peak is not present. A TPD curve of Pd-SSZ-13(Sr) without cobalt might be useful.

Specific comments

-          In Figure 1.A. the name of the sample is Co-SSZ-13(Sr) or Co-SSZ-13(Na)?

-          In Figure 1.A. the difference between spectra with different color should be explained.

-          In Figure 1.C. pink color of the sample seems to be white. Photo with white background would be more convincing.

-          Sign “~” in the text in several sentences: would be better to give its meaning in words or to skip it. (e.g. section 2.1. line 13, and figure capture of Figure 2.)

-          On page 4, the first sentence of the paragraph is missing.

-          On page 5, the last paragraph of section 2.1. is mainly a duplicate of the previous paragraph.

-          In Figure 2. A, C, D and Figure 3. A, B, C: a second Y-Axis for temperature would be needed to correctly present / interpret the TPD curves.

-          It might be useful to show the NO storage capacity of Co-SSZ-13(Na) aged at 850°C (e.g. Supplementary Figure).

-          The TPD curves of Figure 3.B and C could be presented in a single graph.

-          The name of the samples on the figure legends Figure 3.B,C and in the figure captures aren’t the same.

-          In section 2.3, XRPD patterns of the aged samples would be useful to demonstrate the preservation of the zeolite structure.

-          It would be useful to present in a table the NO adsorption capacity at 100°C (mmol/g) for all samples presented in Fig.2. and 3.

-          In Figure 4., the measured conversion vs. temperature values should be plotted, not only the fitted curve.

-          It would be interesting to show the catalytic activity of Cu-SSZ-13(Sr) before ageing.

-          In section 3, line 15: heating rate unit should be corrected: °C / min

-          On page 9 flow rate unit should be corrected.

-          Were the samples before in situ static IR experiments evacuated?

-          On page 9, references for XANES should be put to reference list.

-          Fig.5. and Fig.S1. are the same.

-          Experimental methods in section 3 are repeated in Supplementary Information.

-          Conditions of ageing should be given in section 3.

-          Experimental details of the SCR catalytic test should be given in section 3.

-          On page 10, section 4: This sentence should be amended: “The resulting materials have higher adsorption capacity for NO under different conditions” compared to what?

-          For Ref. [42] the year of publication is not correct.

-          The reference list contains self-citations in a very high ratio.

Reviewer 3 Report

Comments and Suggestions for Authors

This contribution describes the preparation of SSZ-13 with higher amount of Al pairs caused by the Sr(II) cation in the synthesis gel. Through this study, the authors tried to exhibit the merit of this SSZ-13, the excellent hydrothermal stability. Unfortunately, I don’t recommend this article to be published on this journal for some reasons.

1. The article is not logically organized. For example, the amount of Al pairs in the zeolite structure is the key factor in this study, however, the necessary physical data, such as elemental analysis of ICP, XRD, BET and 27 Al solid-state NMR, is not provided. 

2. I agree with that the higher amount of Al pairs lead to a better hydrothermal stability. As mentioned above, the changes in the fresh and aged metal-exchanged SSZ-13 should be the important information for the reviewers and the readers. Although the authors provide the excellent PNA capacity of aged Pd/Co-ssz-13 after hydrothermal aging treatment at 850, 900, and 930 oC, the changes in the catalysts are not mentioned. The authors should provide more information about the phenomenon and the corresponding explanation. 

3. In Figure 4, the 920 oC-aged Cu-SSZ-13 showed excellent SCR activity. Nevertheless, the SCR activity of fresh Cu-SSZ-13 should be included, and the changes in the BET surface area or XRD should be provided as well.

4. In case of the synthesis of SSZ-13, the author should give more evidence of the influence of Sr cation by NMR and XRD. 

Comments on the Quality of English Language

The English is good.

Round 2

Reviewer 1 Report

Comments and Suggestions for Authors

The manuscript has been well improved. 

Author Response

thank you.

Reviewer 2 Report

Comments and Suggestions for Authors

I accept the revised version of the manuscript, but suggest minor corrections:

- In Table1, the headings of the last columns are not readable.

- On page 8, line 9: the numbering of Fig.1B is incorrect.

-On page 9, line 1: Fig.7A is Fig.8A now. 

- No Fig.7 in the revised manuscript.

- On page 10, last line: Fig.1D should be corrected to Fig.8.D.

- In Fig.8.C, the arrow showing the temperature of heating, is missing.

- In Fig.8.D , the upright line at 15 min, should be changed to dotted line.

- Fig.9 might be better to add to Fig.8.C as a new line, to help comparison.

- The data in the table in Fig 17. are hard to read.

- In section 2.3, the numbering of the Figures are not correct in the text. (Fig.3. instead of Fig.10.).

-Fig.11 might be better to add to Fig.10.C as a new line, to help comparison.

Author Response

We addressed all the Reviewers minor editing comments. We fixed the incorrect Fig numbers throughout the text. For better readability, we moved some figures to the Supporting Information. We appreciate the Reviewer's suggestions during the Review process.

Reviewer 3 Report

Comments and Suggestions for Authors

This manuscript can be accepted now.

Author Response

thank you.